# Effects of Cow’s Milk Processing on MicroRNA Levels

**DOI:** 10.3390/foods12152950

**Published:** 2023-08-04

**Authors:** Loubna Abou el qassim, Beatriz Martínez, Ana Rodríguez, Alberto Dávalos, María-Carmen López de las Hazas, Mario Menéndez Miranda, Luis J. Royo

**Affiliations:** 1Servicio Regional de Investigación y Desarrollo Agroalimentario (SERIDA), 33300 Villaviciosa, Spain; mmiranda@serida.org; 2Department of Technology and Biotechnology of Dairy Products, Instituto de Productos Lácteos de Asturias (IPLA-CSIC), 33300 Villaviciosa, Spain; bmf1@ipla.csic.es (B.M.); anarguez@ipla.csic.es (A.R.); 3Laboratory of Epigenetics of Lipid Metabolism, Madrid Institute for Advanced Studies (IMDEA)-Food, CEI UAM+CSIC, 28049 Madrid, Spain; alberto.davalos@imdea.org (A.D.); mcarmen.lopez@imdea.org (M.-C.L.d.l.H.); 4Department of Functional Biology, University of Oviedo, 33006 Oviedo, Spain

**Keywords:** cow’s milk, dairy products, microRNA, biomarkers

## Abstract

MicroRNAs (miRNAs) regulate gene expression and might resist adverse physicochemical conditions, which makes them potential biomarkers. They are being investigated as biomarkers of dairy production systems, based on the variations in their levels in raw milk depending on animal diet and management. Whether miRNA levels can serve as biomarkers for dairy products remains unclear, since technological or culinary treatments, such as fermentation, may alter their levels. Here, 10 cow dairy farms were sampled in Asturias (north-west Spain) and milk samples were subjected to microwave heating or used to produce yogurt or cheese. Total RNA was isolated from raw milk and three derived products, and levels of seven miRNAs, selected based on previous studies as possible milk production system biomarkers, were assessed by RT-qPCR. The treatments decreased levels of all miRNAs to some extent. These results also imply that cheesemaking increases the concentration of miRNAs in this product; raw milk and cheese supposedly may provide similar concentrations of miRNAs, higher than those of yogurt and microwaved milk. They also indicate that the content of certain miRNAs in raw milk cannot necessarily be extrapolated to other dairy products.

## 1. Introduction

MicroRNAs (miRNAs) are non-coding RNAs, only 21–25 nucleotides long, endogenously synthesized, and specific to eukaryotic cells. They are involved in a vast coordination of gene expression regulatory networks. So far, it is known that they mediate post-transcriptional regulation by degrading mRNA or repressing its transcription which results in an attenuation of protein translation [1,2]. miRNAs regulate genes not only in the cells that produce them, but they may regulate genes in other cells too [3]. They have been detected in body fluids such as blood, saliva, and milk [4], which is particularly rich in miRNAs [5]. A comparison between serum and milk miRNAs in humans has concluded that most milk miRNAs are not provided by the blood circulation [6] but originate from their biogenesis in mammary alveolar epithelial cells [7].

In milk, miRNAs are found packaged in vesicles such as milk exosomes and fat globules [8,9]. After milk consumption by human adults, bovine milk exosomes can enhance the miRNA resistance under gastrointestinal digestion and transport to the human colon, and at least some of them transferred to the bloodstream [10,11,12], where they may affect gene expression in humans [13,14]. In other words, milk miRNAs might have bioactive effects in humans, although there are many obstacles and challenges to reach the target tissues [15]. 

The production of miRNAs depends on numerous factors, both within the organism [16] and external in the environment [17,18,19]. For example, miRNA expression differs with milk fraction (fat, whey, and epithelial cells), reflecting differences in several metabolic pathways [20,21]. Characteristics of dairy production systems also influence miRNA profiles and therefore the functional properties of bovine milk [22,23]. The sensitivity of miRNA levels to numerous aspects of animal physiology and farm conditions, coupled with their strong resistance to adverse conditions, including temperature variation, RNase, low pH [24], and even pasteurization [25], and the fact that they can be sampled in a non or minimally invasive manner, make them ideal biomarkers [26]. 

Milk miRNAs vary their levels based on diet and production system. Many studies on miRNAs used as biomarkers have focused on raw milk. However, milk is usually submitted to technological processes before human consumption such as pasteurization, fermentation, or many others. Whether miRNA levels can serve as biomarkers for dairy products remains unclear. Therefore, in the present study, we compared the levels of seven miRNAs in raw milk with the levels in milk subjected to microwave heating, fermentation, or ripening. The concentration of the studied miRNAs in these products was also estimated, for further assay to assess their potential bio-functionality.

## 2. Materials and Methods

### 2.1. Milk Sample Collection and Treatments

Raw tank milk was sampled on 10 dairy farms in Asturias during June and July 2021. The selected farms are included in different production systems (Appendix A).

The 10 samples were transported to the laboratory at 4 °C and then processed the following day. To generate control samples (*n* = 10), 2 mL of QIAzol lysis reagent was added to 1 g of raw milk, and then samples were mixed and stored at −80 °C. To obtain microwaved samples (*n* = 10), 50 mL of raw milk was heated in a 700-W microwave oven for 1 min, and then allowed to cool to room temperature. An aliquot of this milk (1 g) was transferred to a new RNase-free tube, 2 mL of QIAzol lysis reagent was added, and samples were mixed and stored at −80 °C. 

To obtain yogurt samples (*n* = 10), 200 mL of raw milk was pasteurized in a thermostatic bath at 85 °C for 30 min, allowed to cool to 42 °C, and then inoculated with a commercial yogurt starter, which contains *Streptococcus thermophilus* and *Lactobacillus delbrueckii subsp. bulgaricus*, at the dose recommended by the manufacturer (50 units/250 L of milk). Once inoculated, aliquots (100 mL) were transferred into two containers, which were incubated in a water bath at 42 °C until the pH reached 4.5, which occurred after approximately 4 h. Then, an aliquot of yogurt (1 g) was transferred to a RNase-free tube, 2 mL of QIAzol lysis reagent was added, and samples were mixed and stored at −80 °C. The pH and product weight were monitored at all stages.

Cheese samples (*n* = 10) were prepared as described by Hynes et al. [27]. A total of 500 mL of raw milk was inoculated with 10 mL of the starter, prepared according to the dose recommended by the manufacturer (10 units/100 L of milk). The starter culture consisted of mesophilic strains *of Lactococcus lactis subsp. lactis* and *cremoris*. Calcium chloride (1 mL, 20% *w*/*v*, final concentration = 0.02%) was added, the mixture was homogenized by stirring, and then aliquots (200 mL) were transferred into two centrifuge flasks (250 mL volume) and incubated at 26–30 °C for 45 min in a water bath. Then, 65 µL of rennet (Nievi, Bizkaia, Spain; 1 × 10,000) was added, and the milk mixture was allowed to coagulate in a water bath at 30 °C until reaching the appropriate consistency after approximately 90 min. The curd was cut into 5 mm cubes using a sterile stainless steel knife, stirred for 20 min, and centrifuged at 220× *g* at room temperature for 10 min. The entire aqueous phase (whey) was removed, and then 35 mL of saturated brine (NaCl 330 g/L, pH 5.4) was added to curd and kept for 5 min. The mini cheeses were ripened at 10–12 °C in a ripening chamber for one week. After this time, 1 g of cheese was transferred to a Falcon tube, 2 mL of QIAzol lysis reagent was added, and samples were mixed and stored at −80 °C. As during yogurt manufacturing, pH and product weight were monitored at all stages.

### 2.2. Total RNA Extraction and Spike in

Prior to RNA extraction, raw milk and dairy product samples were spiked with defined concentrations of synthetic standard miRNAs. To measure the losses of miRNAs due to different milk processing, 6 fmol of cel-miR-238 (Norgen, Thorold, Canada) was added to the mixture of sample + QIAzol in the case of raw milk, microwaved milk, and yogurt. In the case of cheese, 54 fmol of cel-miR-238 was added to 1 g of cheese in 2 mL of QIAzol, based on our observation (from 10 one-week-old cheeses) that 9 g of milk was necessary to obtain 1 g of cheese. In addition, to compare the amounts of miRNAs between the different products, (1 g raw milk vs. 1 g microwaved milk vs. 1 g yogurt vs. 1 g cheese), 6 fmol of cel-miR-39 (Norgen, Thorold, Canada) was added to the mix (sample + QIAzol) of raw milk and cheese. The use of external synthetic reference miRNAs has been reported since the early work on circulating miRNAs [28].

Total RNA was extracted from aliquots (2 mL) of each mix (sample + QIAzol), which amounted to 40 samples, using the mirVana miRNA isolation kit according to the manufacturer’s instructions. RNA was eluted with 100 μL of RNase-free water. RNA concentration and purity (ratio of absorbance at 260 to 280 nm) were assessed using a Nano-Drop spectrophotometer.

### 2.3. Real Time–Quantitative PCR (RT-qPCR)

Total RNA was used for complementary DNA (cDNA) synthesis using the TaqMan Advanced miRNA cDNA Synthesis Kit, and the resulting cDNA was stored at −20 °C until use. Seven miRNAs were chosen due to their expression level in milk estimated from previous sequencing results [23]. Three miRNAs with high expression levels (more than 190,000 rpm) were chosen: bta-mir-148a, bta-mir-30a5p, and bta-mir-21a5p. Three miRNA with low expression levels (between 150 and 500 rpm) were chosen: bta-mir-451, bta-mir-29b and bta-mir-215. Finally, one miRNA with a limited expression level was chosen: bta-mir-7863. The mature sequence of the miRNA used is shown in Table 1. The levels of these seven miRNAs were determined by RT-qPCR in a StepOne thermocycler The final reaction solution contained 10 μL of 2× TaqMan Fast Advanced Master Mix, 1 μL of 20× TaqMan Advanced miRNA Assay, 4 μL of RNase free water, and 5 μL of 1:10 diluted cDNA. The thermocycler program was set at 95 °C for 20 s, followed by 40 cycles at 95 °C for 1 s, and 60 °C for 20 s. All RT-qPCR reactions were performed in duplicate, and the results were averaged only when the duplicates differed within the 0.5 threshold cycle. To assess miRNA losses due to different milk manipulations, miRNA levels were normalized to those of cel-miR-238, while cel-miR-39 was used to compare the concentration of raw milk and cheese. Then, miRNA levels were estimated using qbase+ 3.1 software and expressed using the △△Ct method in base log^2^ [29].

### 2.4. Prediction of miRNA Structure

Predictions on the secondary structure of selected bovine miRNAs were obtained by using an online application (http://rna.urmc.rochester.edu/RNAstructure.html, accessed on 28 September 2022), using default input conditions. The structures with the lowest free energy of formation (ΔG) were selected for each specific miRNA.

### 2.5. Statistical Analyses

Data were expressed using mean ± standard deviation (SD). Because sample sizes were small and some data showed a skewed distribution based on the Shapiro test, non-parametric statistical analysis was carried out. Pairwise comparisons of miRNA levels among raw milk, heated milk, yogurt, and cheese were performed using the Wilcoxon test for paired data. Significance was defined as *p* ≤ 0.05. All analyses were performed using IBM SPSS Statistics for Windows version 22.0.

## 3. Results

### 3.1. Validation of Milk Treatments

As expected, in yogurt fermentation, a reduction in pH, from 6.63 ± 0.19 to 4.53 ± 0.14, was observed after 4 h of the starter culture addition (Appendix A).

During cheese manufacturing, pH was measured at five timepoints: raw milk, immediately after starter addition, after coagulation, as well as before and after ripening. The pH decreased significantly from the moment the ferment was inoculated. The pH decreased strongly after seven days of ripening. Overall, pH fell from 6.23 to 4.61 (Appendix A), confirming lactic fermentation.

When 206 g of raw milk was used to prepare cheese, the average weight of the fresh cheese, after the removal of whey, was 60.13 ± 7.63 g, and the ripened cheese (after 7 days ripening) weighed 22.26 ± 4.07 g. This means that coagulation and whey removal reduced the weight by approximately 69%, and moisture decrease during ripening reduced weight by an additional 18%. Altogether, cheese yield averaged 11%.

### 3.2. miRNA Losses Due to Milk Processing

Relative levels of bta-miR-148a, bta-miR-21-5p, bta-miR-215, bta-miR-29b, bta-miR-30a-5p, bta-miR-451, and bta-miR-7863, normalized to the levels of spiked cel-miR-238, were compared across raw milk, microwaved milk, yogurt, and ripened cheese (Table 2) to assess the losses of miRNAs after the different treatments. All treatments decreased the levels of all seven miRNAs: approximately a 31% decrease by microwave treatment and yogurt fermentation and approximately a 43% decrease during cheese production. However, not all miRNAs were affected in the same way (Table 3, Figure 1). The reductions after microwave heating varied from 17.20% for bta-miR-30a-5p to 39.42% for bta-miR-451; after yogurt fermentation, from 21.45% for bta-miR-21-5p to 41.62% for bta-miR-451; and after cheese production, from 32.73% for bta-miR-30a-5p to 56.32% for bta-miR-215. The reductions were significant for all seven miRNAs in the case of yogurt and cheese, and only for four of seven miRNAs in the case of microwaving (bta-miR-148a, bta-miR-21-5p, bta-miR-215, and bta-miR-451).

### 3.3. miRNA Concentrations in Milk and Cheese

Relative levels of bta-miR-148a, bta-miR-21-5p, bta-miR-215, bta-miR-29b, bta-miR-30a-5p, bta-miR-451, and bta-miR-7863, normalized to the levels of spiked cel-miR-39, were compared across raw milk and ripened cheese to assess the concentration of miRNAs in raw milk and ripened cheese. The differences in levels between the same amount of raw milk and cheese were not significant for the studied miRNAs, which might indicate that their concentrations were similar between raw milk and cheese (Table 4).

### 3.4. Determination of Secondary Structure of miRNA

The secondary structure of each miRNA was predicted using a bioinformatic tool. We have selected the structures with the lowest free energy of formation (Figure 2). The negative values of ΔG were predicted for bta-miR-29b and bta-miR-30a-5p, indicating that the formation of the secondary structure is most suitable. By contrast, the largest section of the stem-loop structure was in bta-miR-21-5p followed by bta-miR-451.

## 4. Discussion

Here, we evaluated the effects of different cow’s milk treatments on miRNA content in order to guide efforts to define biomarkers for assessing the quality or provenance of dairy products destined for human consumption and also to evaluate the contribution of different dairy products in miRNAs, considering them bioactive compounds [30]. Our results confirmed the presence of miRNAs in the studied dairy products and suggest that thermal treatment of raw milk by the microwave as well as yogurt or cheese production can substantially reduce miRNA levels, indicating that the levels of potential biomarker miRNAs in raw milk cannot be necessarily extrapolated to dairy products derived from that milk. 

Microwave heating has been shown to affect certain physical and chemical characteristics of milk [31] and to damage DNA [32], which led us to hypothesize that this treatment could affect miRNAs. Indeed, we found that microwave treatment of raw milk significantly decreased the amounts of bta-miR-148a, bta-miR-215, bta-miR-21-5p, and bta-miR-451. A previous study also described a significant decrease in bta-miR-21-5p but not bta-miR-29b [33]. We did not observe a significant decrease in bta-miR-29b, yet other work reported a 40% loss in miR-29b [34]. This discrepancy may be due to the storage of raw milk prior to treatment. Howard et al. [34] studied the stability of miRNAs in milk after 15 days of cold storage, being after that heated by microwave (*n* = 3). No significant difference was seen in miR-29b after 15 days of cold storage, but the significant difference appeared after microwave heating [34]. This family of milk miRNA has also been reported as sensitive to high pressure processing (HPP) [35].

Uneven stability of individual miRNAs to milk processing methods has been previously pointed out [35]. The fact that milk treatment significantly affects some miRNAs and not others may depend on the different fractions of milk where the miRNAs are found [21]. Intracellular miRNAs are less stable than extracellular miRNAs within exosomes, microvesicles, apoptotic bodies, high-density lipoproteins, or protein complexes [36]. In the case of milk fat, the fat globule membrane appears to be more resistant to gastrointestinal enzymes [37] and microwave heating [38] than other milk components. 

In contrast to microwave heating, milk pasteurization has been reported not to significantly affect the miRNA content of fat or milk cells [25] or extracellular vesicles in milk [39]. Although the homogenization process can cause a significant loss of miRNAs [34], the secondary structure might influence each miRNA stability.

As cheese and yogurt can be presented in different forms on the market, a simple model of these treatments was elaborated, representing the general characteristics of each product. In contrast to the effect of microwaving, fermentation of previously pasteurized milk to make yogurt significantly reduced levels of many miRNAs [34,40]. The reduction in miRNAs during milk processing is not surprising given the range of changes that occur during milk fermentation to make yogurt and cheese. 

In yogurt production, the starter culture can lower the pH below 4.6, leading to aggregation of caseins [41]. Lactose is converted into lactic acid and several amino acids and fatty acids (especially stearic and oleic) are released into yogurt. During bacterial fermentation, vitamin B content increases and minerals are converted into an ionic form [42]. This study revealed that fermentation of previously pasteurized milk to make yogurt significantly reduced the levels of many miRNAs, with a loss of up to 41.62% in some cases.

We suspect that much of the miRNA loss in our study can be attributed to degradation of exosomes, as others have proposed [34]. In one study, fermentation was found to reduce by 90% the protein content of milk exosomes; assuming that, under the effect of bacterial proteases, exosomes can be altered, consequently the miRNAs contained in these exosomes can be easily degraded [40]. Assuming that pasteurization does not significantly reduce the miRNA content [25], most of the reduction in the miRNA content could be attributed to exosome degradation, as described before.

In our study, we prepared cheese following the modified protocol of Hynes et al. [27], obtaining similar yields to those reported in the original work. We observed a more acidic pH on day 7 (4.61) than in that work (5.21), perhaps because we used raw milk for cheese making, so the natural bacteria in the milk could also metabolize lactose. Pasteurization, in contrast, destroys most bacteria, limiting the acidification [43]. The pH can also vary depending on the starter culture used [44].

As it was expected, the greatest losses of miRNAs occurred during cheese manufacturing, given the losses due to fermentation (as in yogurt) but also because of the removal of whey, which is known to contain a wide variety of miRNAs [20,21].

However, we found that concentrations in the final product of the seven miRNAs (1 g of milk vs. 1 g of cheese, spiked by cel-miR-39) did not differ significantly between raw milk and cheese (Table 4). Similarly, two studies reported even higher miRNA concentrations in two types of cheese (camembert and Fresco queso dip) than in raw milk [34,45]. We suspect that fermentation and whey drainage may reduce absolute miRNA levels, but that the subsequent water loss during ripening increases their concentrations.

Apart from the milk fraction where the miRNAs are found, the differences in miRNA losses in the studied processes could also be attributed to the secondary structure of the individual miRNAs. However, although the stem-loop structure predicted for both *bta*-miR-21-5p and bta-miR-451 could indicate higher stability than the loop structure presented in other miRNA as, for example, bta-miR-148a, bta-miR-215, bta-miR-30a-5p, and 7863 (Figure 2), no association was found between the predicted secondary structure and the decrease in miRNA levels. Although the secondary RNA structure could be relevant in miRNA resistance to degradation, there are other structure aspects that need to be considered, including the content in GC and the specific sequence [46,47]. However, it has been proposed that beyond GC content, changes in sequence, structure, and putative RNase A substrate motifs can impact the stability of dietary small RNAs [47], suggesting, overall, that the stability of dietary miRNAs should be experimentally validated one by one. No differences were also found due to the putative level of expression in raw milk (Table 2 and Table Table 3).

Most of the dairy products with quality labels are processed agro-food products. Previous works have pointed to miRNAs as possible traceability biomarkers in raw cow’s milk because of their sensitivity to farm conditions [22,23] or cow breed [48,49]. Unfortunately, technological processes reduce the amount of miRNAs unevenly, complicating the process of identifying traceability biomarkers for the different dairy products, since not all technological processes reduce miRNAs equally. Further analysis, with larger sampling, should be designed to compare dairy products made from milk produced under different farm managements or breed, for example.

Two milk miRNA we used in this study, bta-mir-21-5p and bta-mir-30a-5p, have been demonstrated to increase their plasma concentration in humans after bovine milk consumption [50], although it remains unknown whether this miRNA concentration might be sufficient to produce gene modulation in the consumer [12]. Since technological processes reduce the amount of miRNAs (Table 3), we can assert that more research is needed on the functionality and bioavailability of miRNAs in each of the different dairy products.

## 5. Conclusions

We confirmed that bovine milk contains several miRNAs even after microwaving, pasteurization followed by fermentation to make yogurt, and cheese manufacturing. We showed that these treatments decreased all miRNA levels.

Our results clearly argue for caution in efforts to identify miRNAs that may be useful biomarkers; they may need to be assessed in the final products for human consumption, and not merely extrapolated from assays of the raw milk from which they are produced.

Finally, considering miRNAs as bioactive components in milk and dairy products, raw milk and cheese supposedly may provide similar concentrations of miRNAs, higher than those of yogurt and microwaved milk, although the miRNA profile may differ between these two products. Additional studies are needed to explore the complete profiles and availability of miRNAs in dairy products and, subsequently, their putative functionality in human cells.

## Figures and Tables

**Figure 1 foods-12-02950-f001:**
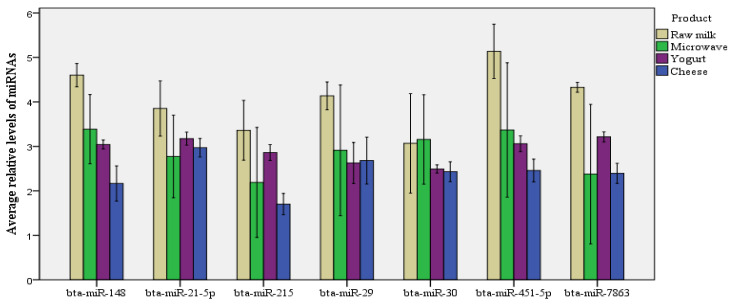
Loss of miRNA content in raw milk after microwave heating, yogurt fermentation, and cheese manufacture. Average relative levels of bta-miR-148a, bta-miR-21-5p, bta-miR-215, bta-miR-29b, bta-miR-30a-5p, bta-miR-451, and bta-miR-7863 in raw milk (*n* = 10), microwave-heated milk (*n* = 10), yogurt (*n* = 10), and cheese (*n* = 10). Levels were normalized to those of spiked cel-miR-238. The bar chart shows the average miRNA level for each product and the standard error bars.

**Figure 2 foods-12-02950-f002:**
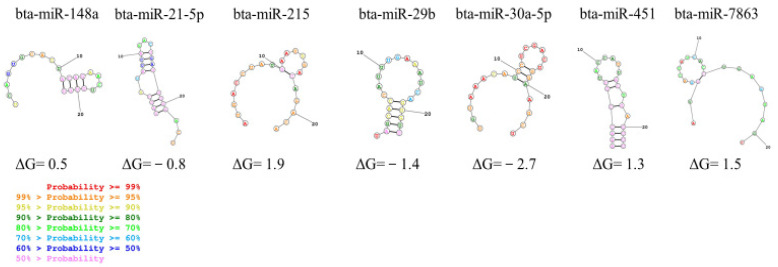
Predicted secondary structures of bovine miRNAs. The motifs with the lowest predicted free energy of formation (ΔG) were selected as the most likely structures.

**Table 1 foods-12-02950-t001:** Mature sequence of miRNA used.

MiRNA	Mature Sequence
bta-mir-148a	UCAGUGCACUACAGAACUUUGU
bta-mir-21-5p	UAGCUUAUCAGACUGAUGUUGA
bta-mir-215	AUGACCUAUGAAUUGACAGACA
bta-mir-29b	UAGCACCAUUUGAAAUCAGUGUU
bta-mir-30a-5p	UGUAAACAUCCUCGACUGGAAGCU
bta-mir-451	AAACCGUUACCAUUACUGAGUU
bta-mir-7863	AUGGACUGUCACCUGAGGAGC
cel-mir-238	UUUGUACUCCGAUGCCAUUCAGA
cel-mir-39	UCACCGGGUGUAAAUCAGCUUG

**Table 2 foods-12-02950-t002:** Relative levels of the seven miRNAs of interest in raw milk, microwave-treated milk, yogurt, and ripen cheese *.

MiRNA	Raw Milk		Microwave		Yogurt		Cheese	
	Mean	SD	Mean	SD	Mean	SD	Mean	SD
bta-miR-148a	4.62	0.72	3.44	0.87	3.04	0.16	2.17	0.63
bta-miR-21-5p	4.04	0.62	2.86	1.08	3.18	0.23	2.97	0.33
bta-miR-215	3.90	0.85	2.60	1.24	2.86	0.28	1.70	0.38
bta-miR-29b	4.20	0.34	2.98	1.64	2.63	0.73	2.68	0.83
bta-miR-30a-5p	3.61	0.91	2.99	0.94	2.49	0.15	2.43	0.35
bta-miR-451	5.24	0.42	3.17	1.99	3.06	0.28	2.46	0.40
bta-miR-7863	4.28	0.39	2.33	1.75	3.21	0.18	2.39	0.36

SD, standard deviation, * Levels were normalized to those of spiked cel-miR-238 (see Section 2.3).

**Table 3 foods-12-02950-t003:** Decrease in miRNA levels after treatment of raw milk *.

miRNA	Microwaved Milk		Yogurt		Cheese	
	Decrease (%)	*p*-Value	Decrease (%)	*p*-Value	Decrease (%)	*p*-Value
bta-miR-148a	25.53	0.043	34.22	0.005	53.15	0.005
bta-miR-21-5p	29.18	0.012	21.45	0.013	26.51	0.005
bta-miR-215	33.20	0.012	26.58	0.013	56.32	0.005
bta-miR-29b	29.07	0.080	37.46	0.028	36.15	0.028
bta-miR-30a-5p	17.20	0.063	31.04	0.013	32.73	0.013
bta-miR-451	39.42	0.018	41.62	0.005	53.09	0.005
bta-miR-7863	45.45	0.123	24.84	0.005	44.01	0.005
Mean	31.29		31.03		43.14	

Decreases are expressed as the percentage in raw milk. Levels of miRNA were normalized to those of spiked cel-miR-238, * *p*-value vs. raw milk (Wilcoxon test).

**Table 4 foods-12-02950-t004:** Relative abundance levels of seven miRNAs in raw milk and cheese *.

MiRNA	Raw Milk		Cheese		*p*-Value (Wilcoxon Test)
	Mean	SD	Mean	SD	
bta-miR-148a	2.32	1.35	2.10	0.30	0.76
bta-miR-21-5p	2.30	2.01	2.97	1.64	0.33
bta-miR-215	1.68	2.06	1.60	0.17	>0.99
bta-miR-29b	1.72	2.26	3.16	1.63	0.11
bta-miR-30a-5p	1.77	1.95	2.89	0.33	0.13
bta-miR-451	2.24	2.58	1.50	0.11	0.31
bta-miR-7863	1.99	1.46	2.43	1.21	0.40

* Levels were normalized to those of spiked cel-miR-39.

## Data Availability

The data that support the findings of this study are available from the corresponding author upon reasonable request.

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
