# Peer review of "Effects of Cow’s Milk Processing on MicroRNA Levels"

_foods, 2023, doi:10.3390/foods12152950_

Round 1
Reviewer 1 Report
1. Besides resistance to adverse physicochemical conditions, are there other important characteristics for biomarkers?
2. Line 42, have any orally ingested miRNAs from milk been found in the circulation system?
3. Line 55, could the author list some examples of biomarker miRNAs?
4. Lines 55-56, as the authors mentioned that milk needs to be processed before it becomes human food. Why the effects of pasteurization were not evaluated? Instead, the effect of microwave heating was assessed.
5. Lines 126-127, why cel-miR-238 and cel-miR-39 were used for normalization and comparison for the concentration of raw milk and cheese, respectively?
6. Statistical analysis needs to be done and labeled in Tables and Figures.
7. Why did the author evaluate the secondary structure of the miRNAs?
8. Please check the punctuation for the entire manuscript.
Please check the punctuation for the entire manuscript.
Author Response
Thank you very much for taking the time to review our work.
Besides resistance to adverse physicochemical conditions, are there other important characteristics for biomarkers?
Of course, there are other characteristics for biomarkers, for example, to be non-invasive, objectively and easily measured, inexpensive, and must give information about a biological state or process.
Apart from that they should have a high sensitivity, allowing early detection, and no overlap in values between problem samples and controls, and should be biologically plausible.
We think that is not necessary to include in the manuscript the characteristics of a biomarker
- Line 42, have any orally ingested miRNAs from milk been found in the circulation system?
Bovine milk miRNA bta-mir-21-5p and bta-mir-30a-5p, have been demonstrated to increase their plasma concentration in humans after bovine milk consumption (Wang et al., 2017, https://doi.org/10.1093/jn/nxx024). This information has been added at the end of Discussion section.
- Line 55, could the author list some examples of biomarker miRNAs?
In this work we don´t focused on single miRNA. The individual miRNA we used in this work were chosen due to their expression level, estimated from sequencing results previously published in Abou el qassim et al., 2022 (https://doi.org/10.3390/vetsci9120661): Three miRNA with high expression levels (between 190.000 rpm and 500000): bta-mir-148a, bta-mir-30a5p and bta-mir-21a5p). Three miRNA with low expression level (between 150 and 500 rpm: bta-mir-451, bta-mir-29b and bta-mir-215). And finally, one miRNA with limited expression level: bta-mir-7863. Examples of biomarkers can be found in Abou el Qassim et al, 2022a and b, https://doi.org/10.3390/ijms231911681, https://doi.org/10.3390/vetsci9120661, and LeGuillou et al 2019 https://doi.org/10.1038/s41598-019-56690-7, for example.
This information has been added at the end of Discussion section
- Lines 55-56, as the authors mentioned that milk needs to be processed before it becomes human food. Why the effects of pasteurization were not evaluated? Instead, the effect of microwave heating was assessed.
The effect of pasteurization was evaluated in a previous work already published in a congress paper (Abou el qassim, L.; Royo, L.J. The effect of pasteurization in the expression of bovine milk microRNA. Book of Abstracts of the 72nd Annual Meeting of the European Federation of Animal Science. 2021, 27, 637) reference 25 of the manuscript, and no significative difference was found between before and after pasteurization. We decided to include microwave heating following other references 34 and 37 in the manuscript. A recent paper (Kim et al, Metabolites, 2023, https://doi.org/10.3390/metabo13020139) also measures the changes in miRNA during microwave heating.
- Lines 126-127, why cel-miR-238 and cel-miR-39 were used for normalization and comparison for the concentration of raw milk and cheese, respectively?
We are measuring the levels of miRNA in tow ways, First the decrease of miRNA due to treatment, normalized using cel-mir-238 in all treatments, and second the final concentration in the dairy product. Only during cheese making there is a volume loss, so that we need a different normalizer, using in this case cel-mir-39.
- Statistical analysis needs to be done and labeled in Tables and Figures.
Statistical analysis was done in all Tables
- Why did the author evaluate the secondary structure of the miRNAs?
Two possible explanations for uneven loss of miRNA were taken into account: milk fraction where the miRNAs are found, or miRNA secondary structure, although no association was found between the predicted secondary structure and the decrease of miRNA levels (lines 282-283).
- Please check the punctuation for the entire manuscript.
Done
Reviewer 2 Report
Dear authors,
the manuscript "Effects of cow's milk processing on microRNA levels" reports information about the presence of miRNAs both in raw milk and in derived products. The article is clear and well written, so I do not have revision request, but, I have a question to the authors. Do you know which could be the human gene targets of these seven MiRNAs? If yes, if it is present in literature, could you please add this point in the discussion section?
Try to revise the english with a native language speaker
Author Response
Thank you very much for taking the time to revise our article.
In relation to the question, we have previously studied the functionality of some of these miRNAs in relation to their level in cow's milk according to the production system, for example for bta-mir-215 in Abou el qassim et al, 2022- Differences in the microRNAs levels of raw milk from dairy cattle raised under extensive or intensive production systems. https://doi.org/10.3390/vetsci9120661. But, as far as we know there is no publication about the functionality of these bovine miRNA in human milk consumers, although it has been demonstrated that the concentration in human plasma of some of the miRNAs tested (bta-mir-21-5p and bta-mir-30a5p) increases after bovine milk consumption (Wang et al, The Journal of Nutrition, 2018, https://doi.org/10.1093/jn/nxx024).
Apart from the potential use of these miRNA as biomarkers for dairy production system, the next step in our work is the study of the different functionality of milk, and may be dairy products, depending on the production system, in line with your comments. A paragraph has been added at the end of discussion section.
Reviewer 3 Report
This is an interesting manuscript aimed to compare the levels of seven miRNAs in raw and treated milk, and to assess their potential functionality as biomarkers. However, it is not clear if the evaluated miRNAs were finally tested and proposed as biomarkers for raw milk or any dairy product derived from milk (i.e., microwaved milk, yogurt and cheese). I also suggest considering next comments:
- Line 18: Replace “and were subjected” by “and milk samples were subjected”.
- Line 21: Replace “biomarker” by “biomarkers”.
- Line 25: A final conclusive sentence related to the objective appears to be missing.
- Lines 65, 79, 89 and 95: Please correct the sign for the temperature value.
- Line 81: Cited reference should be numeric and placed in square brackets.
- Line 135: Why the authors did not include any statistical test to prove the miRNAs as biomarkers for raw milk or dairy products derived from milk?
- Lines 173 and 177: What was the difference between Table 2 and Figure 1; both appear to provide very similar information.
- Lines 205-210: Did the authors suggest that miRNAs are biomarkers in raw milk but not in dairy products derived from milk?, please clarify.
- Line 218: Cited reference should be numeric and placed in square brackets.
- Line 251: Remove “(2000)”.
- Line 258: Separate reference number from the text.
- Line 272: In Conclusions section, it is not clear if the seven miRNAs were finally considered as biomarkers according to the results of this study.
- Line 308: In References section, only the firs letter of the article title should be capitalized.
Author Response
Thank you very much for taking the time to review our work.
The individual miRNA we used in this work were chosen due to their expression level, estimated from sequencing results previously published in Abou el qassim et al., 2022 (https://doi.org/10.3390/vetsci9120661): Three miRNA with high expression levels (between 190.000 rpm and 500000): bta-mir-148a, bta-mir-30a5p and bta-mir-21a5p). Three miRNA with low expression level (between 150 and 500 rpm: bta-mir-451, bta-mir-29b and bta-mir-215). And finally, one miRNA with limited expression level: bta-mir-7863. Taking that into account, miRNAs had not been chosen to be tested as biomarkers, and it has not been done. Anyway, tank milk has been sampled in dairy farms with different production systems, to avoid differences due to farm management. This information has been added in Material and Methods section, item 2.3. A paragraph has been added at the end of discussion section.
Other comments
- Line 18: Replace “and were subjected” by “and milk samples were subjected”. Done
- Line 21: Replace “biomarker” by “biomarkers”. Done
- Line 25: A final conclusive sentence related to the objective appears to be missing.
There is nothing missing in the Abstract. From line 21 to 25 we quote our conclusion
- Lines 65, 79, 89 and 95: Please correct the sign for the temperature value. Done
- Line 81: Cited reference should be numeric and placed in square brackets. Done
- Line 135: Why the authors did not include any statistical test to prove the miRNAs as biomarkers for raw milk or dairy products derived from milk?
We used non-Parametric statistical analysis because sample sizes were small and some data showed a skewed distribution. Pairwise comparisons of miRNA levels were performed using the Wilcoxon test for paired data. With these tests we showed in most cases that there are significant differences between levels of miRNA in milk and dairy products.
In previous works, we have studied the use of miRNAs as biomarkers of milk production (Abou el qassim et al, Veterinary Sciences 2022a y Abou el qassim International Journal of Molecular Sciences 2022b, https://doi.org/10.3390/vetsci9120661, https://doi.org/10.3390/ijms231911681). In this work we wanted to see if we can still use these markers in dairy products, but as we have seen, technological processes reduce miRNAs differently, so this could lead to a large variability in miRNA levels that complicates the use of these molecules as biomarkers even more. Taking all that into account, we are convinced that our sampling is sufficient to compare between milk processes, but not to compare between systems, even though, we used tank milk samples from dairy farms with different productions systems. Furthermore, given the results of this work, the most important issue to keep in mind, is that the results from raw milk miRNA expression levels cannot be directly extrapolated to dairy products.
Further analysis with larger sampling should be designed to compare between dairy products made from milk produced under different farm managements for example.
- Lines 173 and 177: What was the difference between Table 2 and Figure 1; both appear to provide very similar information.
We think that Table 2 and Figure 1 complement each other. Table 2 presents the percentage decrease of miRNAs in the different dairy products refereed to raw milk, but does not compare the products with each other. It also presents the level of significance of this difference. However, Figure 1 shows the variability and the comparison between all products.
- Lines 205-210: Did the authors suggest that miRNAs are biomarkers in raw milk but not in dairy products derived from milk?, please clarify.
In other studies, we have studied the use of miRNAs as biomarkers of milk production systems and according to these studies it seems that the feeding and management system changes the levels of some miRNAs. Here we wanted to see if we can still use these markers in dairy products, but as we have seen, technological processes reduce miRNAs differently, so this could lead to a large variability in miRNA levels, and that variability complicates the use of these molecules as biomarkers even more.
Furthermore, given the results of this work, it is important to keep in mind that the results from raw milk cannot be directly extrapolated to dairy products.
Further works will be design to compare individual miRNA levels among dairy products coming from milk produced under different dairy systems.
- Line 218: Cited reference should be numeric and placed in square brackets. Done
- Line 251: Remove “(2000)”. Done
- Line 258: Separate reference number from the text. Done
- Line 272: In Conclusions section, it is not clear if the seven miRNAs were finally considered as biomarkers according to the results of this study.
The seven miRNA have not been selected because of their use as biomarkers, but as an example of miRNA differentially expressed in raw milk (based on sequencing results)
- Line 308: In References section, only the firs letter of the article title should be capitalized. Done
Reviewer 4 Report
The authors evaluated miRNAs in the raw milk from cattle as associated with various treatments of the milk.
Major issues.
The number of samples employed is small. Also, the authors did not perform a correct stratification of the sampling. The criteria employed for selecting the samples are unclear.
This cannot be rectified at this point, but at least the authors a) should explain their strategy for sampling and b) perform the necessary analysis to identify possible differences between samples in accord to their origin.
I consider the above a significant limitation in this study.
Also, no details of the PCRs are provided. Again, this is a serious omission.
Minor issues
There are some recent relevant references, which can be included in the discussion in order to better explain the findings.
Author Response
Thank you very much for taking the time to review our work.
Major issues.
The number of samples employed is small. Also, the authors did not perform a correct stratification of the sampling. The criteria employed for selecting the samples are unclear.
In our work we want to know to what extent milk processing affects miRNA content. Similar articles studying the effect of milk processing on miRNA content used the same number of samples, ten milk samples (Kim et al, Metabolites 2023, https://doi.org/10.3390/metabo13020139) or even less, as few as three (Howard et al, J Agric Food Chem. 2015, https://doi.org/10.1021/jf505526w; Smyczynska et al., PlosOne 2020, https://doi.org/10.1371/journal.pone.0236126). We used milk samples from commercial farms in the region, and as we know that the production system affects miRNA content (Abou el qassim et al, Veterinary Sciences 2022a y Abou el qassim International Journal of Molecular Sciences 2022b, https://doi.org/10.3390/vetsci9120661, https://doi.org/10.3390/ijms231911681) we sampled tank milk from farms with different management systems, shown in Supplementary Table 1. Furthermore, analysis was done with statistical tests adapted for that (Wilcoxon test).
This cannot be rectified at this point, but at least the authors a) should explain their strategy for sampling and b) perform the necessary analysis to identify possible differences between samples in accord to their origin.
I consider the above a significant limitation in this study.
In this work we wanted to see if we can still use these markers in dairy products, but as we have seen, technological processes reduce miRNAs differently, so this could lead to a large variability in miRNA levels that complicates the use of these molecules as biomarkers even more. Taking all that into account, we are convinced that our sampling is sufficient to compare between milk processes, but not to compare between systems, even though, we used tank milk samples from dairy farms with different productions systems. Furthermore, given the results of this work, the most important issue to keep in mind, is that the results from raw milk miRNA expression levels cannot be directly extrapolated to dairy products.
Further analysis with larger sampling should be designed to compare between dairy products made from milk produced under different farm managements for example.
In previous works, we and others have demonstrated that farm management, including diet, exercise, affects milk miRNA levels, so that, our sampling include tank milk from different dairy production systems, intended to cover sufficient variability, but not to compare between systems. Therefore, by comparing 10 samples of raw milk from farms of various production systems (with the intention of randomising this factor) with their respective dairy products, we have seen that the different treatments affect the miRNAs and that these are affected differently (some are less stable than others depending on the treatment). Therefore, this adds more variability which complicates the task of using these miRNAs as biomarkers.
Also, no details of the PCRs are provided. Again, this is a serious omission.
We dedicate section 2.3 to describe the PCR, the results of PCR can be seen in the Figure 1, showing the average relative levels of the studied miRNA in the different products, where the variation was also reflected.
We use, as in previous works, the TaqMan™ Advanced miRNA Assay (thermofisher, https://www.thermofisher.com/order/catalog/product/A25576), and all related products, cDNA synthesis kit (TaqMan Advanced miRNA cDNA Synthesis Kit, A28007), and TaqMan Fast Advanced Master Mix (4444557). These products are supplied and already tested by thermofisher to be sensitive and specific, as well as for PCR efficiency and wide dynamic range. Anyway, three out of seven miRNA used in this study has been tested in our laboratory by means of serial dilutions, finding correct PCR efficiencies (from 90% to 104%) and R2 (from 0,94 to 1). All miRNA tested (seven bta-mir and two cel-mir) were purchased from thermofisher, either inventoried (bta-mir-148a, bta-mir-21a-5p, bta-mir-29b, bta-mir-451, cel-mir-238, cel-mir-39) or custom-made (bta-mir-215, bta-mir-30a-5p, bta-mir-7863). We cannot provide primers and probes used for the amplification, because are included in the commercial TaqMan™ Advanced miRNA Assay. To solve that question, we have included in the manuscript a new table (Table 1) with the mature sequence of the nine miRNAs used, in section 2.3.
Minor issues
There are some recent relevant references, which can be included in the discussion in order to better explain the findings.
Some new references and two new paragraphs have been added at the end of discussion section.
Round 2
Reviewer 4 Report
The authors have improved the manuscript.